# Recent Advances of the BIRALET System about Space Debris Detection

**Tonino Pisanu** [1,*] , **Giacomo Muntoni** [2] , **Luca Schirru** [1] , **Pierluigi Ortu** [1] , **Enrico Urru** [3] and **Giorgio Montisci** [1,2]

1 National Institute for Astrophysics (INAF), Cagliari Astronomical Observatory, Via della Scienza 5, 09047 Selargius, Italy; luca.schirru@inaf.it (L.S.); pierluigi.ortu@inaf.it (P.O.); giorgio.montisci@unica.it (G.M.)
2 Department of Electrical and Electronic Engineering, University of Cagliari, Piazza D'Armi snc, 09123 Cagliari, Italy; giacomo.muntoni@unica.it
3 Italian Space Agency, Via del Politecnico, 09133 Rome, Italy; enrico.urru@asi.it
* Correspondence: tonino.pisanu@inaf.it; Tel.: +39-070-71180237

**Abstract:** Space debris is internationally recognized as a planetary threat. Efforts to enhance the worldwide radar monitoring networks have been intensified in the last years. Among the new radars employed for the observations, one of the most promising is the Bistatic Radar for Low Earth Orbit (LEO) Tracking (BIRALET), which employs the Sardinia Radio Telescope as a receiving segment. The Sardinia Radio Telescope (SRT) has recently been proven to be a reliable instrument for space debris monitoring and, for this purpose, over the years has undergone some substantial modifications in order to be able to rise to the status of a fully functional radar receiver. However, an extensive measurement campaign, in order to assess the real potential of the radar, has never been done before. In this paper, the authors present the first real space debris measurement campaign of the SRT, made between December 2018 and October 2019 using the new dedicated channel of the P-band receiver. A total of 27 objects were correctly detected during this campaign, characterized by a radar cross section (RCS) interval between 0.13 and 13.4 m$^2$ and a range interval between 459 and 1224 km.

**Keywords:** Sardinia Radio Telescope; bistatic radar; space debris

## 1. Introduction

Based on recent estimates, the number of orbital debris in space amounts to 139 million [1]. The overpopulation of objects within the space environment increases the chance of several events that could lead to catastrophic consequences, such as collisions, fragmentations, and reentries in the Earth's atmosphere [2,3]. Such a risk forced worldwide space agencies to develop a set of countermeasures to prevent and/or mitigate possible threats. The latter take the form of the space situational awareness (SSA) program and, more specifically, of one of its constituent segments: the space surveillance and tracking (SST), which focuses on the active and inactive objects that orbit the earth [4]. The embodiment of this planetary commitment can be figured out paying attention to the size of the huge network of sensors dedicated to space monitoring [5–7]. Despite the undeniable supremacy of the United States, the necessity of an organized SST segment is also prioritized by other countries [7].

Within this framework, recently, Italy has also been involved in the European monitoring program for SST. The two main radar systems for space debris monitoring in Italy are the BIRALES (Bistatic Radar for Low Earth Orbit Survey) [8] and the BIRALET (Bistatic Radar for Low Earth Orbit Tracking) [9] (refer to the Supplementary Materials). These bistatic radars share the same transmitter, but, for the first system, the receiver is the Northern Cross, located in Medicina, near Bologna (Northern Italy), whereas for the second one, the receiver is the Sardinia Radio Telescope (SRT), located near Cagliari (Sardinia, South-West Italy) [9,10]. Starting from early 2014 [11], the SRT has been involved

in the Italian and European space debris monitoring program and it has been gradually upgraded to make it more suitable for space debris observations [12]. As pointed out in both [11,13], the first measurements performed in April 2014 were planned to assess the potential of the SRT as a receiver in a bistatic radar system for space debris monitoring. On that occasion, the observation schedule included a total of six pieces of debris, each one correctly spotted [11]. However, these first results, although encouraging, showed the shortcomings of the employed system, tied also to the inadequacy of the back-end (e.g., speed of the frequency sweep tied to the resolution bandwidth selected, poor processing capabilities, and missing information due to the sampling time), which was mainly used for the primary mission of the radiotelescope, i.e., radioastronomical observations [9,14].

Because of these limitations, in a recent paper, the authors presented the design and realization of a new space debris dedicated channel for the SRT [12,13]. The new channel is based on the introduction of a few filtering stages and a down-conversion section necessary to match the working frequency of the new back-end, based on the FPGA commercial board Red Pitaya (model STEMLab 125-14: https://www.redpitaya.com/f130/STEMlab-board, accessed on 19 March 2021). The channel has been tested taking advantage of the reentry of the Chinese space station Tiangong–1 on April 2018, in a joint operation with the BIRALES system, with the aim to retrieve the speed of the falling object from the Doppler measurements [13]. However, as the Tiangong-1 was characterized by an overall radar cross section (RCS) of about 20 m$^2$, that measurement cannot be considered a real testing ground for the capabilities of the new system. Moreover, the observation of the Tiangong–1 was focused on the calculation of the speed from the Doppler frequency of the space station. Therefore, a further set of observations has been scheduled, including objects with a rather wide interval in terms of RCS and range. Accounting for objects with different RCS and range is pivotal for an extensive campaign to assess the capabilities of a radar system. Small and far objects can give a hint on the sensitivity of the system, whereas large and near objects are useful for the calibration of the receiving chain, as the blinding of the receiver must be avoided. Despite the great distance between radar and debris, the reflected signal could be high enough to saturate the amplification stage of the receiving chain.

In this paper, the most extensive space debris measurement campaign ever performed by the BIRALET system is presented. Previous measurement sessions reported too few detections to be considered a solid benchmark for future campaigns. The measurements discussed herein were taken in a kind of *beam-parking* mode: the objects were part of a predetermined list provided by the Italian Air Force. This compromise was necessary, as the transmitter is located in a military facility. Unfortunately, this also means that we had no control of the pool of observable objects. For every object in the list, the SRT was pointed at the right coordinates (predicted by means of two-line element sets—TLE [15]), waiting for the passage of the object inside the beam. During the campaign, performed between December 2018 and October 2019, a total of 27 objects were correctly detected, with a RCS variation in the range of 0.13–13.4 m$^2$ and a slant range (SR—intended as the sum of the ranges of the object from the receiver and the transmitter) variation between 918 and 2077 km. The results show also that the estimated Doppler shift was predicted with a maximum error of about 1 kHz.

## 2. The BIRALET System

The BIRALET system is a bistatic radar configuration located in Sardinia (Italy). The transmitter of the BIRALET is the TRF (Trasmettitore a Radio Frequenza—Radio Frequncy Transmitter), a 7 m fully steerable wheel-and-track parabolic antenna with a primary focus configuration, located in the region "Salto di Quirra" (Northeast from Cagliari). The azimuth and elevation range of the transmitting antenna are 0–360 deg and 0–90 deg, respectively, with an accuracy of 0.1 deg and a speed of 3 deg/s in both directions [8]. The TRF is capable of transmitting a continuous wave (CW) signal of 10 kW at 410 MHz. The receiver of the BIRALET system is the SRT, a 64 m fully steerable wheel-and-track parabolic antenna (Figure 1) devoted mainly to radio astronomical observations, located in

Pranu Sanguini (Northeast from Cagliari), operating in the frequency range from 0.3 to 116 GHz. In addition to the 64 m primary parabolic mirror, the antenna is equipped with a 7.9 m secondary mirror and the beam waveguide system (BWG), which includes two 2.9 m mirrors and one 3.9 m mirror, for a total of four focal positions [16–20]. To better clarify the mirrors and foci configuration of the SRT, a sketch of the antenna is reported in Figure 2, whereas Table 1 summarizes the key features of the radio telescope. One of the peculiarities of the SRT is the active surface system that allows the modification of the profile of the primary mirror (by means of an electro-mechanical control) in order to compensate for deformations tied to gravitational load, pressure of the wind, and thermal gradients [21].

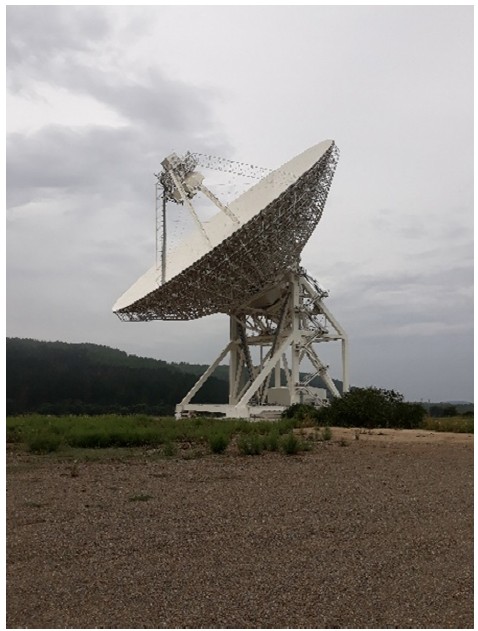

**Figure 1.** Photo of the Sardinia Radio Telescope (SRT).

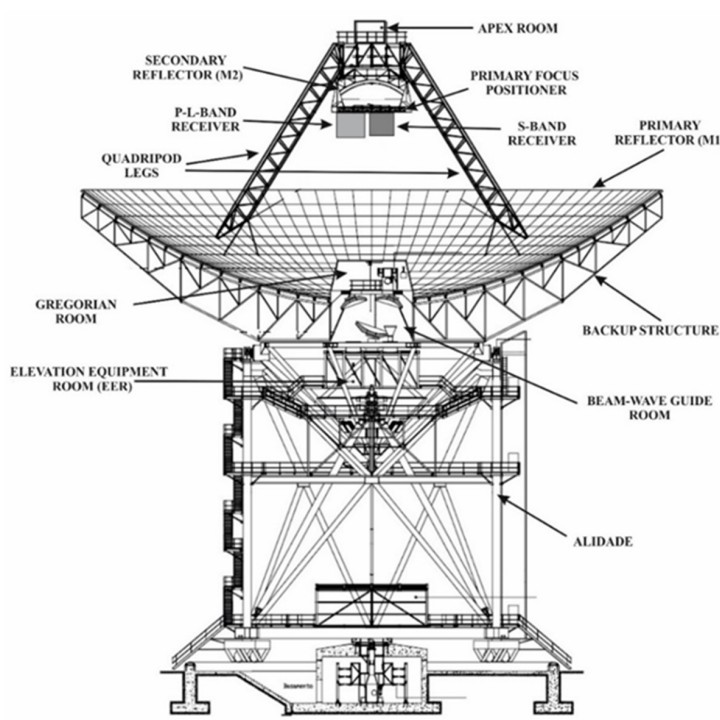

**Figure 2.** Sketch of the Sardinia Radio Telescope.

**Table 1.** Key features of the Sardinia Radio Telescope (SRT). BWG, beam waveguide system.

| Optics | Gregorian (Shaped) + BWG |
| --- | --- |
| Focal Positions | Primary: f/D = 0.33<br>Gregorian: f/D = 2.34<br>2 × BWG I: f/D = 1.38<br>2 × BWG II: f/D = 2.81 |
| Frequency Range | 0.3–116 GHz |
| Primary Reflector Diameter | 64 m |
| Secondary Reflector Diameter | 7.9 m |
| BWG Mirrors Diameter | 2.9–3.9 m |
| Azimuth and Elevation Speed (wind speed < 60 km/h) | 0.85 °/s (Az)<br>0.5 °/s (El) |
| Antenna Gain at 410 MHz | 46.6 dBi |
| Antenna Efficiency at 410 MHz | 57.7% |

The sensitivity of the bistatic system was computed in terms of scattered power from the targets, using the well-known radar equation [22] for the calculation of the received power and Equation (1) for the evaluation of the noise floor:

$$N_F = k_B T_{sys} B_N \tag{1}$$

where $k_B$ is the Boltzmann's constant; $T_{sys}$ is the system temperature (as sum of the antenna and receiver temperature); and $B_N$ is the noise bandwidth, evaluated as the bandwidth of the output filter of the receiver. Figure 3 shows the simulated performances of the BIRALET system in terms of received power versus RCS for the range of values from 200 to 2000 km. The estimated signal-to-noise ratio (SNR) for a specific RCS can be derived easily by the difference between the received power and the noise floor. It is worth mentioning that the latter represents an ideal estimation of the SNR received, because, in a real measurement, the noise floor tends to be higher.

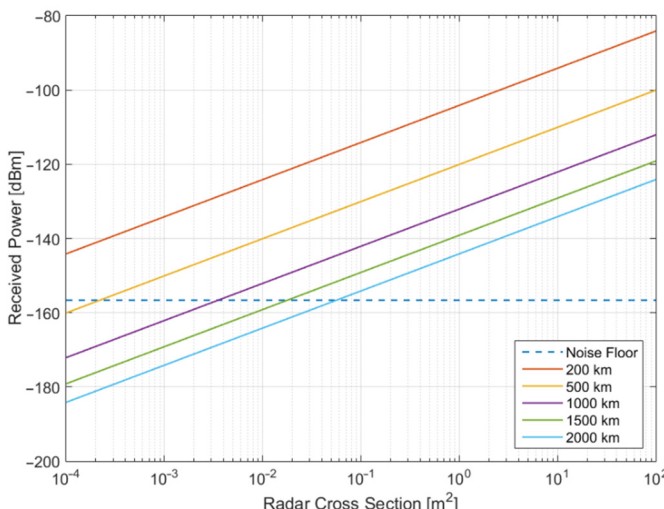

**Figure 3.** Simulated received power at the antenna (SRT) as a function of the radar cross section (RCS) and range of the object for the Bistatic Radar for Low Earth Orbit (LEO) Tracking (BIRALET) system.

The receiver employed for the space debris observations is the P-band receiver, working between 305 and 410 MHz [13,23]. The receiver block is composed of five main sections: the coaxial feed, the cryogenic front-end, the linear to circular polarizer, the noise calibrator and antenna unit injection, and the filter selector. The radio frequency (RF) signal is received by the coaxial feed and sent to the cryogenic front end, directly connected to the

noise calibration and antenna unit injection, for the calibration of the entire system. Then, the signal is sent to the polarizer, which allows the retrieval of the signal in both linear and circular polarizations. The last stage is the choice of a suitable filter depending on the type of measurement. More details on the P-band receiver can be found in [23]. The half-power beamwidth (HPBW) of the SRT in P-band (410 MHz) is about 0.8 deg, as shown in the radiation pattern reported in Figure 4.

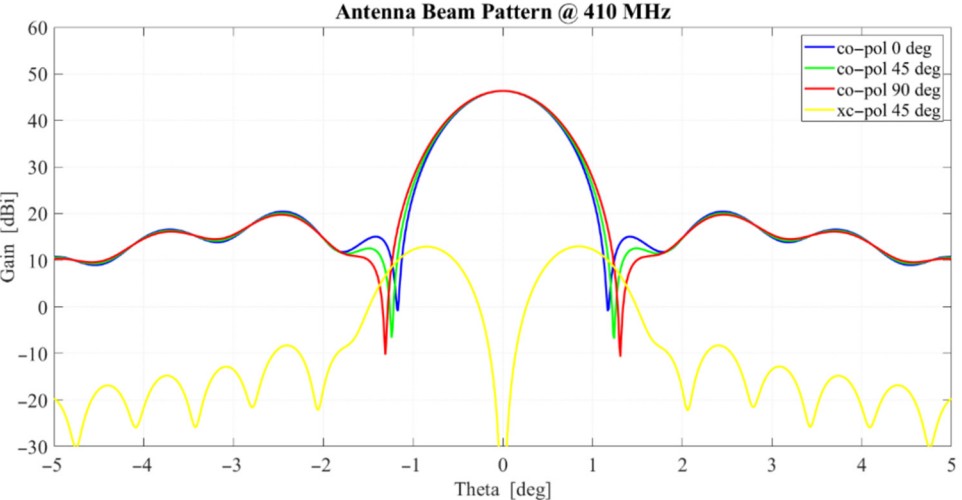

**Figure 4.** Radiation pattern of the SRT in P-band (410 MHz).

The P-band receiver block is the first part of the space debris dedicated channel of the SRT. A complete schematic of the channel is reported in Figure 5. After the P-band receiver block, the signal goes in input to the focus selector, which allows the choice of the focus for the observation (the primary focus in our case). This section is characterized by a set of filters, amplifiers, and attenuators, providing an overall gain of about 20 dB. From the top of the parabolic reflector, the signal is then carried down to a shielded room at ground level, by means of a 500 m optical link. Inside the shielded room, downstream to the optical link, the signal passes through the IF distributor, which allocates the signals coming from the antenna to the different back-ends, and is characterized by an overall gain of about 20 dB.

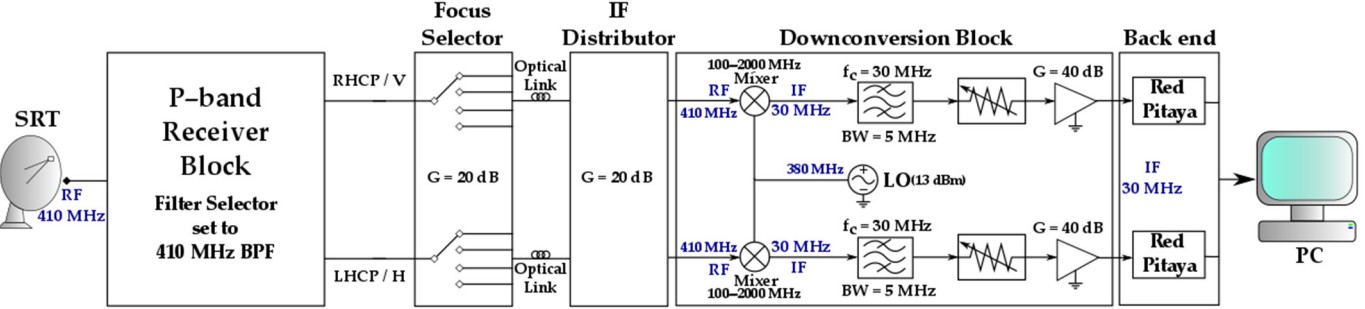

**Figure 5.** Schematic of the space debris dedicated channel of the SRT. RF, radio frequency.

The dedicated back-end for space debris observations is composed of the down-conversion block, and, finally, the Red Pitaya. The down-conversion block lowers the 410 MHz frequency of the incoming signal to match the working frequency of the back-end (0–50 MHz). The down-converted signal is filtered and amplified. A digital step attenuator, controlled by the Red Pitaya, is located upstream to the final amplifier in order to avoid saturation of the system. The setting of the digital attenuator is calibrated upon the power level of the carrier, as explained in [13]. The last block of the dedicated channel is the Red Pitaya, a commercial system-on-a-chip FPGA board provided with a dual-core 800 MHz ARM9 processor, a FPGA with 28,000 logic cells, 512 MB of DDR3 RAM system memory,

2 RF inputs and 2 RF outputs, an overall 3 dB-bandwidth of 50 MHz, a sample rate of 125 M Sample/s, and 14 bit of ADC resolution. A digital signal processing chain was built inside the board, allowing the conversion of the input signal in a base-band complex signal. This raw signal can be stored and post-processed using an FFT engine and, finally, displayed on a computer.

## 3. Results and Discussion

Between 13 December 2018 and 10 October 2019, the BIRALET system was able to observe a total of 27 objects. The Italian Air Force (Aeronautica Militare—AM) has provided the list of objects to observe, making sure to include a wide variety of samples in terms of RCS and range. All the observations were made in a variant of the beam-park mode. Usually, the beam-park mode (or stare mode) is employed for the survey of space debris, illuminating a specific portion of the sky and waiting for the objects to cross the field of view (FOV) of the antenna. It is employed very often to detect objects that are not part of a catalogue. In this case, the aim was not to detect uncatalogued objects, but to assess the capabilities of the radar system. Furthermore, the characteristics of the BIRALET system are not suitable for survey purposes, owing to the narrow beam. Thus, the antenna was pointed each time in a different direction, dictated by a specific object in the list, and waited for its passage. Before the observations, a forecasting simulation was made in order to obtain the azimuth and elevation pointing coordinates and to estimate the Doppler shift values. An algorithm based on the reading of the TLEs provides these pieces of information. The results of the campaign are summarized in Figures 6 and 7, representing the RCS as a function of the range and the altitude as a function of the elevation pointing angle, respectively, for the 27 detected objects.

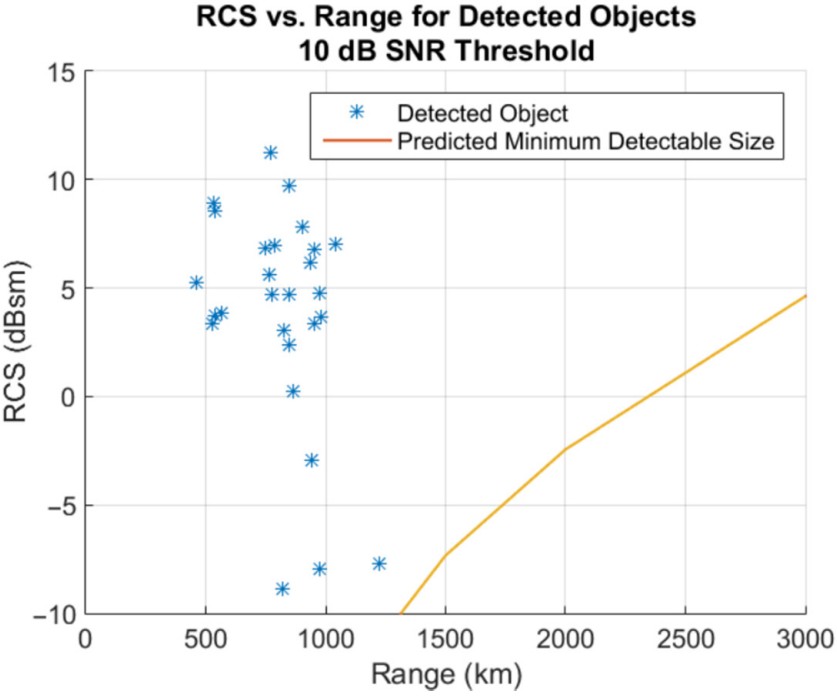

**Figure 6.** Scatter plot of RCS vs. range for the detected objects with respect to the minimum detectable size by the BIRALET system. SNR, signal-to-noise ratio.

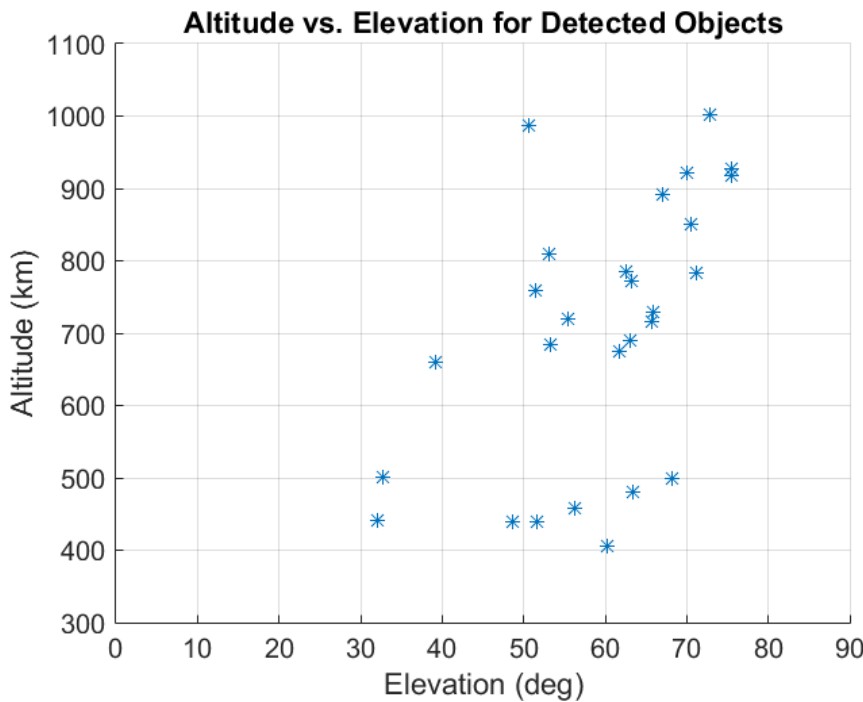

**Figure 7.** Scatter plot of altitude vs. elevation for the detected objects.

As expected, with reference to the scatter plot in Figure 6, the detected objects are located above the curve, representing the predicted minimum detectable size by the BI-RALET system. This threshold value was obtained accounting for a minimum SNR of 10 dB (see Figure 3). It is worth noting that the SNR depends on many factors such as rotation of the object, materials, uncertainty in the values of the RCS and trajectory provided by the TLEs, pointing accuracy, and possible unwanted losses in the receiving chain. Moreover, especially in our case, where the transmitter is handled by a third part, the actual SNR can be difficult to estimate owing to unknown fluctuations of the transmitted power. For the above reasons, in this work, SNR was employed only for a first estimate of the system ability to detect a particular object.

The RCS of the detected objects varies between 0.13 m$^2$ (−8.86 dBsm) and 13.4 m$^2$ (11.27 dBsm), whereas the range varies between 459 and 1224 km. However, the majority of detections are clustered between 500 and 1000 km. From this image, it is clear that the BIRALET system would be capable of detecting even smaller objects, especially for low range values. Notwithstanding, the object with an RCS of 0.13 m$^2$ is the smallest object ever detected by the bistatic radar, so far. During the reception of the scattered signals from the larger and/or closer objects, the receiving system did not suffer from saturation problems, thanks to an accurate setting of the digital step attenuator. From Figure 7, it can be inferred that all the spotted objects are located in the low Earth orbit (LEO), which includes altitudes between 200 and 2000 km. Detections on higher orbits are not feasible, owing to the relatively low power of the transmitter. However, almost every radar space debris sensor is focused on LEO measurements, as it is the most populated one.

As mentioned in Section 2, the data recorded by the back-end are acquired in a row format that can be later post-processed. These refined data are displayed in a spectrogram fashion, showing the duration of the passage, frequency, and power. An example of a time–frequency spectrogram is reported in Figure 8, for the object 4256, which is characterized by an RCS of 0.51 m$^2$, an altitude of 918.1 km, and a range of about 941.5 km.

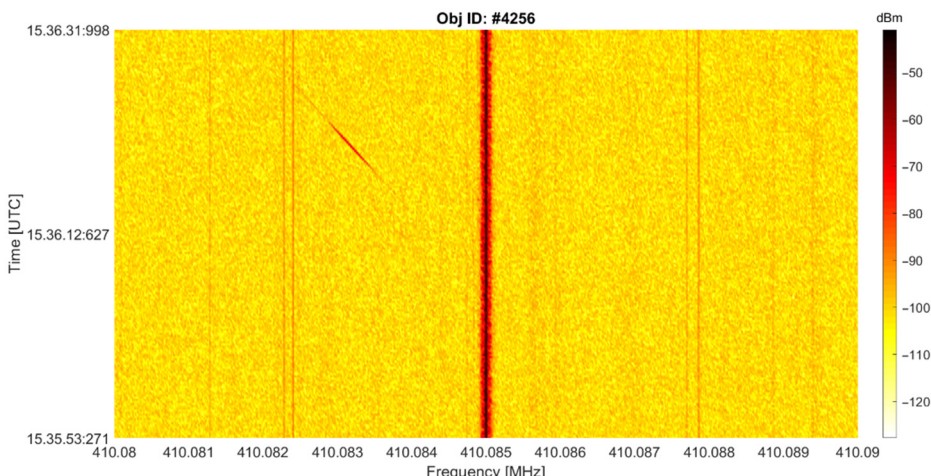

**Figure 8.** Spectrogram of the passage of the object 4256, registered on 19 May 2019.

The passage of the objects within the beam of SRT is clearly recognizable thanks to its sweep in time and frequency. The thick line in the middle of the diagram (centered at 410.85 MHz) represents the carrier signal. Its presence is due to the relative proximity of the receiver with the transmitter of the BIRALET (the baseline of the bistatic radar is about 40 km). Actually, this peculiarity simplifies the interpretation of the measured Doppler frequency, which can be calculated from the frequency shift between the object echo and the carrier at the center of the visibility interval (which corresponds to 15:36:20 UTC in the given example). The measured Doppler frequency was retrieved for all of the objects and compared with the estimated one, showing a good agreement between the observed and predicted Doppler frequency. The maximum difference is in the order of magnitude of 1 kHz, the mean value of the difference is about 0.34 kHz, and the standard deviation is about 0.27 kHz. This comparison is reported in Table 2, for every object in the list, identifiable by its catalogue number (NORAD ID).

**Table 2.** Estimated and measured Doppler shift for the 27 objects detected.

| NORAD ID | Estimated Doppler (kHz) | Measured Doppler (kHz) | Error—Absolute Value (kHz) |
|---|---|---|---|
| 26102 | −7.96 | −7.44 | 0.52 |
| 36119 | 5.78 | 5.62 | 0.16 |
| 39194 | 4.22 | 4.44 | 0.22 |
| 15494 | 7.28 | 7.26 | 0.02 |
| 39452 | −8.66 | −8.82 | 0.16 |
| 39453 | −9.43 | −9.75 | 0.32 |
| 33492 | 8.49 | 8.16 | 0.33 |
| 30774 | −1.59 | −2.11 | 0.52 |
| 28931 | −5.85 | −4.66 | 1.19 |
| 3047 | −7.19 | −7.28 | 0.09 |
| 26222 | −3.63 | −3.49 | 0.14 |
| 27944 | 7.12 | 6.88 | 0.24 |
| 8458 | 3.53 | 3.40 | 0.13 |
| 6350 | 2.66 | 2.07 | 0.59 |
| 15595 | 7.80 | 7.42 | 0.38 |
| 36508 | 9.21 | 9.60 | 0.39 |
| 25482 | −6.31 | −6.48 | 0.17 |
| 11166 | −5.01 | −4.28 | 0.73 |
| 25105 | −7.69 | −8.04 | 0.35 |
| 4256 | −1.70 | −1.65 | 0.05 |
| 1328 | −4.04 | −3.30 | 0.74 |
| 7363 | −6.64 | −6.66 | 0.02 |
| 7337 | 4.87 | 4.90 | 0.03 |
| 22824 | 5.25 | 5.54 | 0.29 |
| 40961 | 9.59 | 9.92 | 0.33 |
| 12150 | −1.18 | −0.87 | 0.31 |
| 7646 | 4.64 | 5.36 | 0.72 |

## 4. Conclusions and Future Works

In the last six years, the BIRALET system has been actively involved in the European space debris monitoring program, following an upgrade schedule that culminated in the

installation of a dedicated receiving channel. Although the capabilities of the newly added channel have already been assessed in a recent paper [13], an extensive measurement campaign to test the real potential of the radar system was still lacking. In this paper, the most substantial radar measurement campaign for the BIRALET system is presented. Based on a list of objects provided by the Italian Air Force, the BIRALET has been able to detect a total of 27 objects with RCS ranging from 0.13 to 13.4 m$^2$ in a range interval between 459 and 1224 km. In the future, this campaign will stand as a solid foundation for the next important step in the evolution of the BIRALET for space debris monitoring purposes: the tracking system. This new feature will enable the possibility to employ the BIRALET system for orbit determination of known and unknown objects in both single and joint campaigns with other Italian and European radar sensors.

**Supplementary Materials:** The following are available online at https://www.mdpi.com/2226-431 0/8/3/86/s1.

**Author Contributions:** Conceptualization, T.P., G.M. (Giacomo Muntoni), and L.S.; methodology, T.P., G.M. (Giacomo Muntoni), L.S., and P.O.; validation, T.P., L.S., and P.O.; formal analysis, T.P., G.M. (Giacomo Muntoni), and G.M. (Giorgio Montisci); data curation, T.P., G.M. (Giacomo Muntoni), and G.M. (Giorgio Montisci); writing—review and editing, T.P., G.M. (Giacomo Muntoni), G.M. (Giorgio Montisci), and E.U.; supervision, G.M. (Giorgio Montisci). All authors have read and agreed to the published version of the manuscript.

**Funding:** This work was founded in part by the European Commission Framework Programme H2020 and Copernicus Space Surveillance and Tracking under grants 785257-2-3SST2016 and 237/G/GRO/COPE/16/8935-1SST2016.

**Institutional Review Board Statement:** Not applicable.

**Informed Consent Statement:** Not applicable.

**Data Availability Statement:** Data sharing not applicable.

**Conflicts of Interest:** The authors declare no conflict of interest.

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
