# Peer review of "Recent Advances of the BIRALET System about Space Debris Detection"

_aerospace, doi:10.3390/aerospace8030086_

Round 1

Reviewer 1 Report

Recent Advances of the BIRALET System About Space Debris Detection

Corresponding Author: Tonino Pisanu

Overall Comments:

The authors of this manuscript describe a validation campaign for the BIRALET radar system specifically related to the detection of space debris.  The BIRALET system is described and data are presented from 27 objects that were detected during a campaign that took place in 2014.

I found this manuscript to be clear, concise, and well organized. The system and experiment are well described. My main point of contention with publishing this manuscript is that I found a compelling motivation for publishing this work to be lacking.  This manuscript describes how the BIRALET system detected 27 objects within its expected detection ability when provided with the exact location of the objects prior to the measurement. This study provides no data on the measured detection limits of the system. All targets that were used in this study were well within the detectable region of the system as defined by a 10 dB SNR limit, so it appears that no effort was made to find the limits of the noise floor or the minimum detection capabilities of the system. Lines 171-172 state that “Furthermore, the characteristics of the BIRALET system are not suitable for survey purposes, due [to] the narrow beam”, making it clear that this system will not be used for space debris surveys, however, no description of future work for this system is described so it is unclear if the system is meant for monitoring of space debris that are already being tracked by other radar systems or for some other purpose.  The manuscript is also very similar in content to reference 11: Figure 3 in this manuscript is nearly identical to Figure 2 in reference 11, and Figure 8 in this manuscript appears to be a subset of figures presented in reference 11. This manuscript adds perhaps more validation points that may have not been previously reported, but no new conclusions are reached by this manuscript, so it is unclear to me how this adds to the body of knowledge already present in the scientific literature. It would be useful for the system to track objects near or below the theoretical detection limit to understand the true experimentally determined noise floor and detection capabilities.  Are the actual noise floor and SNRs for the detected objects as expected from simulations?  My opinion is that further characterization of the BIRALET system’s capabilities that have not been reported elsewhere are necessary to make this manuscript ready for publication. My specific comments and edits are included below. 

Line 10: More details are needed in the abstract to orient the reader to the content of the study. Please include the dates of the campaign and quantitative summary conclusions of the study.  BIRALET should be described in the abstract since it is included in the manuscript’s title.

Line 22: I suggest changing “entails” to “increases the chances of”.

Line 30: I suggest deleting “the” prior to “space monitoring”.

Line 32: I suggest changing “also Italy has” to “Italy has also”.

Line 33: I suggest deleting the commas before and after “in Italy”.

Line 35: Please provide a summary reference for BIRALET here where BIRALET is first defined, similar to how reference 8 was provided for BIRALES.

Lines 36 and 38: If a regional location is included for Northern Cross (i.e., “Northern Italy”) please also include a regional location for Cagliari for consistency.

Line 43: I suggest changing “In” to “On”.

Line 44: I suggest deleting “definitively”.

Line 45: Please provide further details for what is meant by “tied also to the inadequacy of the back-end.”  What are the specific inadequacies being referred to?

Line 52: Please provide a weblink and model number for the Red Pitaya board being used.

Line 55: “RCS” should be defined here at first use.  It is currently defined on line 74.

Line 63: By “avoiding the eventuality to “blind” the receiver” do the authors mean “blinding of the receiver must be avoided”?

Line 67: I suggest changing “measurements” to “measurement”.

Line 75: Please describe what is meant by “Slant Range” here as the values of “between 918 and 2077 km” do not agree with the range values of “between 459 and 1224 km” described elsewhere in the manuscript (e.g., line 192).

Line 76: Please include quantitative results for the accuracy of the Doppler shift measurements in place of the qualitative phrase “good accuracy”.

Line 79: Please define the acronym “TRF”.

Line 82: The authors state the speed of the BIRALET antenna to be 3 deg/sec here, but Table 1 indicates the speeds are 0.85 deg/sec for the azimuth and 0.5 deg/sec for the elevation.  Please reconcile this perceived difference.

Line 92: I suggest changing “to modify” to “modification of”.

Line 118: I suggest adding “Simulated” before “received” in the caption of Figure 3.

Line 127: I suggest changing “to retrieve” to “retrieval of”.

Line 128: I suggest changing “polarization” to “polarizations”.

Line 149: I suggest changing “allows to lower” to “lowers”.

Line 153: I suggest deleting “the” prior to “saturation”.

Line 161: I suggest deleting the comma before “and” and replacing “in” with “on”.

Line 163: Line 73 states that the measurement campaign occurred into August of 2019, yet the date of October 10th, 2019 is provided in line 163.  Please reconcile the difference.

Line 172: I suggest changing “due the narrow beam” to “due to the narrow beam”.

Line 199: If LEO is going to be spelled as “Low Earth Orbit” I suggest first spelling it out when it is first used, i.e. at line 34.  I also suggest changing “belong to the” to “are located in”.

Line 204: I suggest changing “This” to “These”.

Line 207: Please provide quantitative properties for object 4256, such as the altitude, range, and RCS.

Line 217: Figure 8 indicates times around 15:36 UTC yet the manuscript on this line indicates a time of 13:36:20 UTC.  Please reconcile the discrepancy.

Line 219: Please include quantitative results for the accuracy of the Doppler shift measurements in place of the qualitative phrase “very good accuracy”.  Lines 76 and 219 indicate “good accuracy” and “very good accuracy”, respectively, for measurements of Doppler frequency compared to estimated values, but quantitative values should be provided.  For example, a column should be added to Table 2 indicating the percent difference between the estimated and measured values. The authors could then average these values to provide a quantitative value for how close the Doppler frequency measurements are in general to the estimated values.

Line 229: Please provide a specific reference for the “recent paper” described on this line.

Reviewer 2 Report

In Section 3, Results and Discussion: I have made some minor points on the manuscript itself, attached here. In addition, I would like to comment on text in Lines 218 and 219:

Do the comparisons really show that the predictions are good, or do they give an indication of the accuracy of the measured Doppler shift? Are you assuming that the observed measurements are perfect?

Perhaps better to say 'showing a good agreement between observed and predicted Doppler frequency'. Also I suggest doing some statistics on the differences given in Table 2 - mean and st deviation of Measured - Estimated Doppler.

Round 2

Reviewer 1 Report

I thank the authors for their diligence in addressing each of my individual comments and concerns and for making suggested changes to the revised version of their manuscript.  I also thank the authors for pointing out the differences between their manuscript and reference 11 that I missed during my initial review.  I disagree with the authors’ comments that “reporting expected Signal-to-Noise Ratio for space debris is ambiguous and not useful.”  I understand that SNR depends on many factors, yet it seems to me that there must be some metric, if not SNR, that could be used to evaluate if the system is reaching expected theoretical performance levels and am surprised at the lack of comparisons with simulated results. Perhaps this information is included elsewhere and I appreciate the authors responding to my suggestion to add a citation to a description of the BIRALET system (reference 9).  However, I could not locate this reference through an internet search making it difficult to further evaluate this system. I defer to the editors for their decision on publication of this manuscript.

Author Response

Dear reviewer, 

we would like to thank you for the time spent helping us improving our manuscript. 

We are sorry that you were not able to locate the reference 9. For your convenience, we attach it with this submission, however reference 9 is about deep space applications, not about space debris detection. If it could be of any help to you, reference 13 is an excellent summary of the BIRALET characteristics, at least for the P-band channel (which is indeed the same used in the presented measurements). 

We would also like to clarify our previous comment about the SNR. We did not meant it in an absolute way. Of course, the SNR is a useful parameter for the evaluation of a system performance. Unfortunately, the SNR depends on many factors such as rotation of the object, materials, uncertainty in the values of the RCS and trajectory provided by the TLEs, pointing accuracy, possible unwanted losses in the receiving chain. Moreover, especially in our case where the transmitter is handled by a third part, the actual SNR can be difficult to estimate due to unknown fluctuations of the transmitted power. For the above reasons, in this work, SNR has been employed only to a first estimate of the system ability to detect a particular object.

On the other side, we would like to point out that the evaluation of the SNR is not constraining in our specific case. In fact, the BIRALET system is a Doppler radar (despite the fact that a modification on the system to perform also range measurements is already in place). In this perspective, our concern is whether the object has been correctly spotted or not, and, of course, how much is the Doppler shift. The latter does not change depending on the SNR value. This is the reason why we have reported the comparison between simulated and measured Doppler frequency (see Table 2). Therefore, we believe that the evaluation of the performance of the system (being a Doppler radar) is contained in Table 2. 

In order to better clarify the above points we have added in the revised paper (between lines 205 and 211) the following sentence:

"It is worth noting that the SNR depends on many factors such as rotation of the object, materials, uncertainty in the values of the RCS provided by the TLEs, pointing accuracy, possible unwanted losses in the receiving chain. Moreover, especially in our case where the transmitter is handled by a third part, the actual SNR can be difficult to estimate due to unknown fluctuations of the transmitted power. For the above reasons, in this work, SNR has been employed only to a first estimate of the system ability to detect a particular object."

However, we agree with the reviewer that the SNR might be a useful parameter to evaluate, for instance, the accuracy of the pointing of a system, or, in some specific case studies, the tumbling of an object. 
For this reason, we are planning a calibration campaign based of the observation of known objects with large RCS, for a better estimation of the SNR values. 

Reviewer 2 Report

Thank you for considering my suggestions. I am now satisfied that my suggestions have been carried through to the manuscript. I recommend that the paper now be published.

Author Response

We would like to thank you for the time spent helping us improving the  manuscript